# Evolutionary analyses of intrinsically disordered regions reveal widespread signals of conservation

**Marc D. Singleton**[1]*, **Michael B. Eisen**[1,2]

**1** Howard Hughes Medical Institute, UC Berkeley, Berkeley, California, United States of America,
**2** Department of Molecular and Cell Biology, UC Berkeley, Berkeley, California, United States of America

* marcsingleton@berkeley.edu

## Abstract

Intrinsically disordered regions (IDRs) are segments of proteins without stable three-dimensional structures. As this flexibility allows them to interact with diverse binding partners, IDRs play key roles in cell signaling and gene expression. Despite the prevalence and importance of IDRs in eukaryotic proteomes and various biological processes, associating them with specific molecular functions remains a significant challenge due to their high rates of sequence evolution. However, by comparing the observed values of various IDR-associated properties against those generated under a simulated model of evolution, a recent study found most IDRs across the entire yeast proteome contain conserved features. Furthermore, it showed clusters of IDRs with common "evolutionary signatures," *i.e.* patterns of conserved features, were associated with specific biological functions. To determine if similar patterns of conservation are found in the IDRs of other systems, in this work we applied a series of phylogenetic models to over 7,500 orthologous IDRs identified in the *Drosophila* genome to dissect the forces driving their evolution. By comparing models of constrained and unconstrained continuous trait evolution using the Brownian motion and Ornstein-Uhlenbeck models, respectively, we identified signals of widespread constraint, indicating conservation of distributed features is mechanism of IDR evolution common to multiple biological systems. In contrast to the previous study in yeast, however, we observed limited evidence of IDR clusters with specific biological functions, which suggests a more complex relationship between evolutionary constraints and function in the IDRs of multicellular organisms.

**Data Availability Statement:** The code used to produce the results and analyses is available on GitHub at https://github.com/marcsingleton/IDR_evolution2023. The following Python libraries were

## Author summary

Proteins are the molecular machines that carry out many processes required for life at an atomic level. Though many proteins use fixed structures to perform their functions, proteins with flexible segments are widespread, especially in multicellular organisms. Furthermore, these intrinsically disordered regions (IDRs) are often involved in essential cellular functions. However, the sequences of IDRs evolve quickly, which challenges traditional

used: matplotlib [69], NumPy [70], pandas [71], SciPy [72], and scikit-learn [73]. Relevant output files, including the estimated parameters of the substitution, BM, and OU models, are available on Zenodo (DOI: 10.5281/zenodo.10308885). There are no primary data associated with this manuscript. All primary data are available from publicly accessible sources described in their corresponding sections.

**Funding:** This work was supported by a Howard Hughes Medical Institute Investigator Award to MBE. The funders had no role in study design, data collection and analysis, decision to publish, or preparation of the manuscript.

**Competing interests:** I have read the journal's policy and the authors of this manuscript have the following competing interests: MBE is a founder and former member of the board of directors of PLOS.

bioinformatics methods that depend on sequence conservation to predict function. Several studies have demonstrated that distributed biophysical features of IDRs are constrained rather than their exact sequences, and a recent study in yeast found that IDRs with common patterns of conserved features were associated with specific functions. Therefore, in this work we ask if IDRs in fruit flies, another common laboratory organism, also have patterns of conservation with associated functions. We build on the previous study by integrating their approach into a fully statistical framework based on mathematical models of trait evolution. Though we identify widespread signals of conservation in the IDRs of fruit flies, we find less evidence of a simple relationship between features and function. These methods and results will provide a valuable resource that can guide future experimental analyses of IDRs in fruit flies and other organisms.

## Introduction

Intrinsically disordered regions (IDRs) are segments of proteins which lack stable three-dimensional structures and instead exist as ensembles of rapidly interconverting conformations [1]. As a result of this structural heterogeneity, IDRs can interact with diverse binding partners. Often these interactions have high specificity but moderate affinity, which permits the efficient propagation of signals by rapid binding and dissociation [2,3]. Furthermore, as IDRs readily expose their polypeptide chains, they are enriched in recognition motifs for post-translational modifications which allow environmental or physiological conditions to modulate their interactions. Accordingly, IDRs often act as the "hubs" of complex signaling networks by integrating signals from diverse pathways and coordinating interactions [4,5]. However, as IDRs are ubiquitous in eukaryotic proteomes, with estimates of the fractions of disordered residues in the human, mouse, and fruit fly proteomes ranging between 22 and 24% [6,7], they are involved in diverse processes [8] including transcriptional regulation [9] and the formation of biomolecular condensates [10].

Despite the prevalence and importance of IDRs in eukaryotic proteomes, associating them with specific molecular functions or biological processes remains a significant challenge. The sequences of IDRs are generally poorly conserved, so traditional bioinformatics approaches which rely on the conservation of amino acid sequences to identify homologous proteins and transfer annotations between them are largely unsuccessful when applied to IDRs. However, several recent studies have demonstrated evidence that IDRs are constrained to preserve "distributed features" such as flexibility [11], chemical composition [12], net charge [13], or charge distribution [14,15]. Because many sequences can yield a region with a specific composition, for example, this mode of constraint uncouples an IDR's fitness from its strict sequence of amino acids. Furthermore, in contrast to folded regions whose precise contacts and packing geometries are easily disrupted by amino acid substitutions, distributed features are robust to such changes, as individual residues only weakly contribute to a region's fitness. For example, a mutation at one site in an IDR that changes its net charge is easily reversed by subsequent compensatory mutations elsewhere in the region. Thus, under this model the sequences of IDRs can rapidly diverge and still preserve their structural or functional properties.

This form of selective constraint can also describe the evolution of more "localized" features in IDRs such as short linear motifs (SLiMs). Because SLiMs are composed of fewer than 12 residues, they form limited interfaces that frequently mediate the transient binding events involved in signaling pathways [16]. Accordingly, they are highly enriched in IDRs, which provide an accessible and flexible scaffold for these interactions [17,18]. While some SLiMs in

IDRs are strongly conserved at specific positions, these constitute a small fraction of disordered residues, estimated at roughly 17% in the yeast proteome [19]. Instead, as SLiMs are compact and often highly degenerate at some positions, they can arise *de novo* from a small number of mutations and therefore have high rates of turnover. Furthermore, when IDRs contain multiple copies of a motif that jointly mediate a high-avidity interaction [16] or a graded response to a signal via the accumulation of multiple phosphorylations [3,20], the individual motifs are under weak selective constraints. As a result, though SLiMs are encoded by specific sequences, in some contexts they may evolve as distributed features that characterize IDRs as a whole rather than specific sites within them [21].

The initial studies demonstrating evidence of constraint were generally restricted to specific features or proteins. However, by comparing the observed values of various IDR-associated properties against those generated under a simulated model of evolution, Zarin *et al.* [22] showed most IDRs across the entire yeast proteome contain conserved features. Furthermore, they identified clusters of IDRs with common "evolutionary signatures," *i.e.* patterns of conserved features, which were associated with specific biological functions. This analysis for the first time provided a global view of the relationship between sequence and function in IDRs. A follow-up study then expanded on this initial finding by applying techniques from machine learning and statistics to predict the functions of individual IDRs using their evolutionary signatures [23].

However, no known subsequent studies have determined if similar patterns of conservation are found in the IDRs of other systems. As another foundational model organism with abundant genomic information across many evolutionary lineages [24–27], the fruit fly, *Drosophila melanogaster*, is a natural choice for subsequent investigation. Furthermore, given its complex multicellular development process and shared signaling pathways with humans, the findings of such a study would significantly advance our understanding of the role of IDRs in gene regulation as well as human health and disease. The concordance of these results with the previously identified IDR clusters would also have profound implications for the broader mechanisms of IDR evolution. For example, the absence of global patterns of evolutionary signatures across IDRs in *Drosophila* would suggest they are property of IDRs which is unique to yeast. In contrast, the identification of clusters similar to those in yeast would support the existence of a taxonomy of IDRs which is conserved across the tree of life. The latter result would represent a significant step towards the creation of resources for the classification of IDRs analogous to those for folded domains such as Pfam [28], CATH [29], or SCOP [30,31].

Therefore, in this work we applied a series of phylogenetic models to dissect of evolution of a set of orthologous IDRs identified in the *Drosophila* genome. Our analyses span multiple levels, ranging from the sequences that compose these regions to the distributed features that characterize them as a whole. For the latter, though the previous approach relied on simulations to generate the null distribution for a hypothesis of no constraint, we instead leveraged a fully statistical phylogenetic comparative framework [32]. By comparing models of constrained and unconstrained continuous trait evolution, *i.e.* the Brownian motion and Ornstein-Uhlenbeck models, respectively, we can demonstrate evidence of selective constraint on features independent of any assumptions about the underlying process of sequence evolution. However, we also propose hybrid approaches that combine simulations with phylogenetic comparative methods to test increasingly refined models of IDR evolution. We found that IDRs exhibit unique patterns of amino acid substitution and that in some proteins disorder itself is a dynamically evolving property. Furthermore, though IDRs are broadly unconstrained along several axes of feature evolution, we identified signals of widespread constraint in IDRs, indicating conservation of distributed features is mechanism of IDR evolution common to multiple biological systems. Unlike in yeast, though, we observed limited evidence of IDR

clusters with specific biological functions, which suggests a more complex relationship between evolutionary constraints and function in the IDRs of multicellular organisms. These conclusions, however, are tempered by several methodological limitations, *e.g.*, the application of continuous models of trait evolution to discrete data, which are explored in greater detail in the discussion.

## Results

### IDRs are shorter and more divergent than non-disordered regions

As many IDRs evolve rapidly, a major challenge for proteome-wide comparative analyses is correctly inferring and aligning homologous IDRs. We therefore relied on a set of over 8,500 high quality alignments of full-length single copy orthologs from 33 species in the *Drosophila* genus which we had previously generated and characterized [33]. (The use of full-length single copy orthologs ensures that IDRs are properly aligned by "anchoring" them to more conserved regions and that the proteins are unlikely to have undergone functional divergence as a result of gene duplication.) We then identified regions with high levels of inferred intrinsic disorder using the disorder predictor AUCPreD [34]. To highlight the unique features of IDR evolution in subsequent analyses, we also extracted a complementary set of regions with low levels of inferred disorder.

Both sets were filtered on several criteria, including the lengths of their sequences and their phylogenetic diversity, which yielded 11,445 and 14,927 regions, respectively, from 8,466 unique alignments. In the subsequent discussion, we refer to these sets as the "disorder" and "order" regions, respectively. To investigate the differences in basic sequence statistics between the two region sets, we first generated histograms from the average length of each region (S1 Fig). Although both distributions span several orders of magnitude, the order regions are generally longer than the disorder regions, with means of 245 and 105 residues, respectively. We then quantified the sequence divergence in each region by fitting phylogenetic trees to the alignments using amino acid and indel substitution models, which are probabilistic descriptions of sequence evolution that are parameterized in terms of the rates of change between residues or between aligned residues and gaps, respectively [35]. The average rates of substitution are significantly larger in the disorder regions, demonstrating that while both sets contain conserved and divergent regions, IDRs are enriched in more rapidly evolving sequences (S1 Fig). We also searched the database of Pfam domains against the full-length *D. melanogaster* proteins in these alignments and measured their overlap with the disorder and order regions. The results show the disorder set has a clear enrichment in regions with no or only small amounts of overlap relative to the order set, indicating that, compared to structured domains, IDRs are especially resistant to homology-based methods of functional annotation (S2 Fig).

### IDRs have distinct patterns of residue substitution

To gain insight into the substitution patterns of amino acid residues in the disorder and order regions, we fit substitution models to meta-alignments sampled from the respective regions. As these models are parameterized in terms of the one-way rates of change from one residue to another, the rates are not necessarily equal for a given pair when the initial and target residues are swapped. For example, the rate of change of valine to tryptophan can be distinct from that of tryptophan to valine. In practice, however, substitution models are typically constrained to fulfill a condition called time-reversibility, as this converts a difficult multivariate optimization of the tree's branch lengths into a series of simpler univariate optimizations [36]. A common method for fulfilling this condition is parameterizing the model in terms of a frequency vector, $\pi$, and an exchangeability matrix, $S$. The frequency vector determines the model's

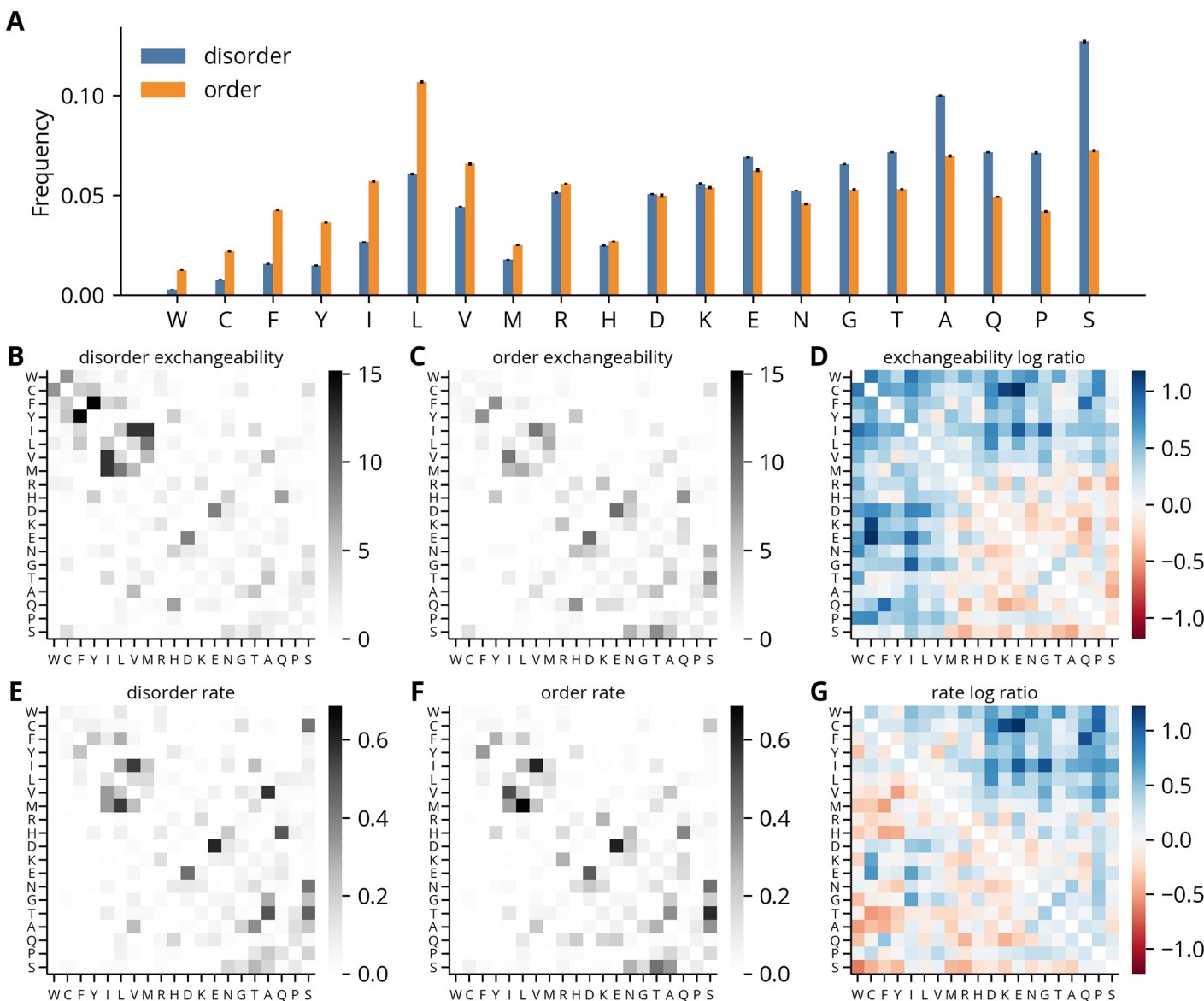

**Fig 1. Amino acid substitution models fit to disorder and order regions.** (**A**) Amino acid frequencies of substitution models. Amino acid symbols are ordered by their enrichment in disorder regions, calculated as the disorder-to-order ratio of their frequencies. Error bars represent standard deviations over models fit to different meta-alignments ($n = 25$). (**B-C**) Exchangeability coefficients of disorder and order regions, respectively, averaged over meta-alignments. (**D**) $\log_{10}$ disorder-to-order ratios of exchangeability coefficients. (**E-F**) Rate coefficients of disorder and order regions, respectively, averaged over meta-alignments. The vertical and horizontal axes indicate the initial and target amino acids, respectively. (**G**) $\log_{10}$ disorder-to-order ratios of rate coefficients.

expected residue frequencies at equilibrium, meaning the model dictates that all sequences eventually approach this distribution, no matter their initial composition. The exchangeability matrix is symmetric ($s_{ij} = s_{ji}$) and encodes the propensity for two residues to interconvert. Because the rate of change from residue $i$ to residue $j$ is given by $r_{ij} = s_{ij} \pi_j$, higher exchangeability coefficients yield higher rates of conversion. Thus, exchangeability coefficients are frequently interpreted as a measure of biochemical similarity between residues.

To highlight the differences in patterns of residue substitution between the disorder and order regions, the parameters in each model are directly compared in Fig 1, beginning with

the frequency vectors. The disorder regions show an enrichment of "disorder-promoting" residues such as serine, proline, and alanine, and a depletion of hydrophobic and bulky residues such as trytophan and phenylalanine (Fig 1A). The exchangeability matrices fit to the disorder and order regions have similar overall patterns of high and low coefficients (Fig 1B–1C). However, the log ratios of the disorder to the order exchangeability coefficients show clear differences within and between the disorder-enriched and -depleted residues. The disorder-enriched residues are less exchangeable with each other, whereas disorder-depleted residues are more exchangeable with each other and with disorder-enriched residues (Fig 1D). Likewise, we observe a trend in the log ratios of the rate coefficients where the coefficients above the diagonal are generally positive, and those below the diagonal are generally negative (Fig 1E–1F). As the coefficients model the one-way rates of substitution between residues with the vertical and horizontal axes indicating the initial and target residues, respectively, this suggests a net flux towards a more disorder-like composition. Though, the coefficients between the disorder-depleted and -enriched classes of residues for both the exchangeability and rate matrices should be interpreted with caution, as they are estimated with a high amount of uncertainty (S6–S7 Figs). A second, more general, caveat is these trends may result from fitting substitution matrices to alignments which were created using other substitution matrices derived from alignments of largely structured proteins. However, as addressed in the discussion, their consistency with several similar analyses suggests they represent true differences in the substitution patterns of IDRs [37,38].

## Intrinsic disorder is poorly conserved in some proteins

Though the substitution models reveal specific patterns of evolution at the level of individual residues, the large amounts of sequence divergence between many orthologous IDRs implies their evolution is not well-described by fine-scale models of residue substitution. Given the growing evidence that IDRs are constrained to conserve distributed properties, we instead turned towards characterizing their evolution in terms of 82 disorder-associated "molecular features" obtained from the previous study of IDRs in the yeast proteome. However, before conducting an in-depth analysis of these features, we examined the disorder score traces in greater detail and were struck by the significant variability between species. For each residue in the input sequence, AUCPreD returns a score between 0 and 1 where higher values indicate higher confidence in a prediction of intrinsic disorder. In some alignments, the disorder scores vary by nearly this entire range at a given position even when there is a relatively high level of sequence identity (Fig 2A). Though we cannot fully eliminate the possibility that this result is a prediction artifact, the strong performance of AUCPreD in a recent assessment of disorder predictors suggests the observed variability reflects true changes in these residues' propensity for disorder [39].

To better understand the relationship of this variability to differences in the regions' biophysical properties, we sought to correlate the average disorder score of the segments in a region with their molecular features. However, as the sequences are not independent but instead related by a hierarchical structure which reflects their evolutionary relationships, any features derived from them are unsuitable for direct use in many standard statistical procedures. In the most severe cases, traits derived from clades of closely-related species can effectively act as duplicate observations, which can yield spurious correlations. We therefore applied the method of contrasts to both the disorder scores and the features [40,41]. This algorithm takes differences between adjacent nodes in the phylogenetic tree relating the species to generate "contrasts," which, under some general assumptions of the underlying evolutionary process, are independent and identically-distributed and therefore appropriate for use in

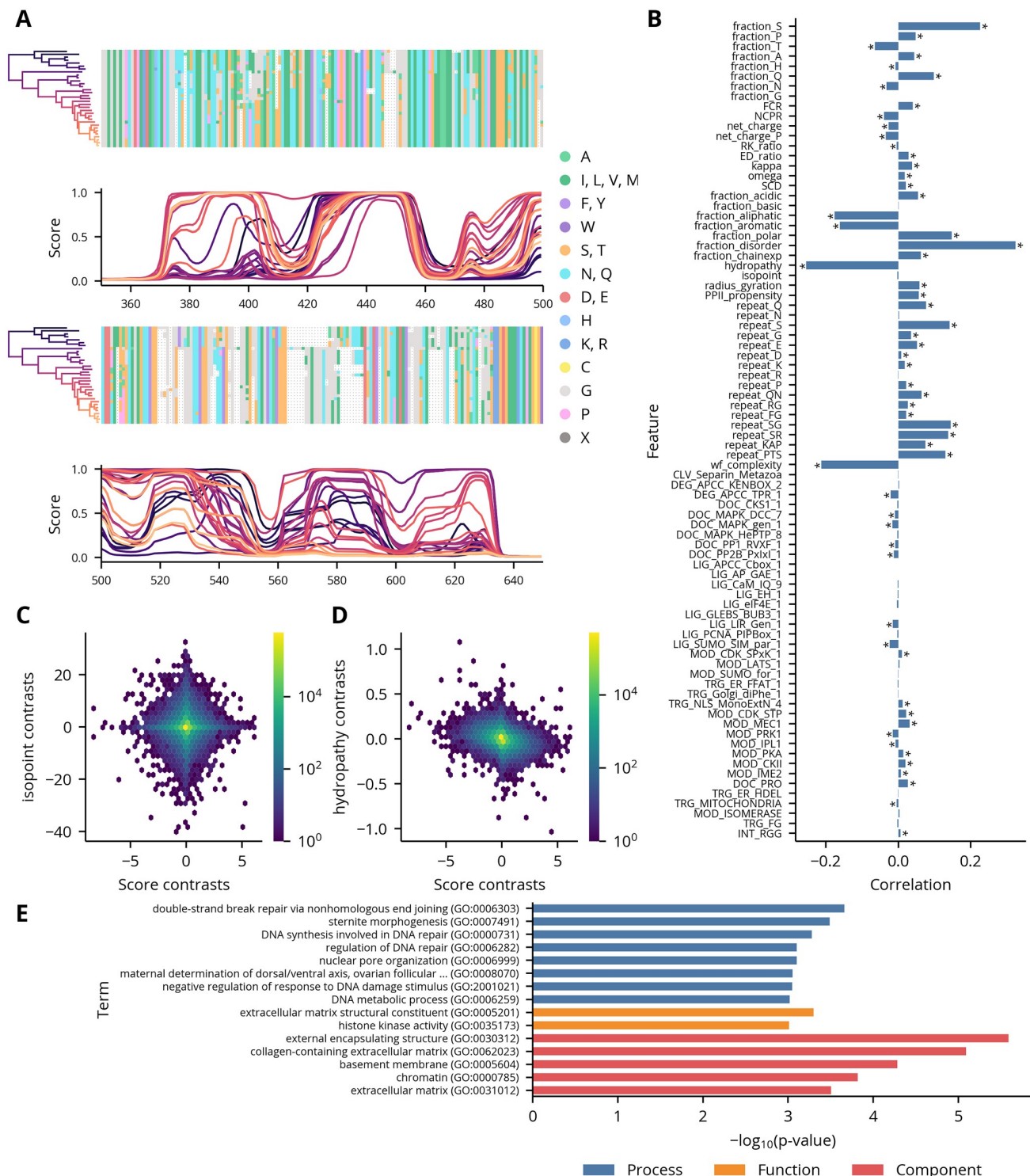

**Fig 2. Analyses of disorder scores.** (**A**) Example region in the alignment of the sequences in orthologous group 07E3 with their corresponding disorder scores. Higher scores indicate a higher probability of intrinsic disorder. Disorder score traces are colored by the position of their associated species on the phylogenetic tree. (**B**) Correlations between disorder scores and feature contrasts in regions. Asterisks indicate statistically significant correlations as computed by permutation tests ($p < 0.001$). (**C-D**) Example scatter plots showing correlations given in panel B. (**E**) GO term analysis of regions with rapidly evolving disorder scores. Only terms where $p < 0.001$ are shown.

correlation analyses. (A sample calculation using this algorithm is shown in S8 Fig.) The resulting feature contrasts have varying degrees of correlation with the score contrasts (Fig 2B–2C). Some, like isopoint, are uncorrelated, but most are significantly, if weakly, correlated. In general, the strongest correlations are observed for features which have a direct biophysical relationship to the presence or absence of disorder, such as disorder_fraction or hydrophobicity. Interestingly, the correlations with many motifs were statistically significant, though small in magnitude relative to the non-motif features. However, a more detailed analysis of this observation is presented in the discussion. To determine if regions with rapidly evolving disorder scores are associated with particular functions, processes, or compartments, we then extracted the regions in the upper decile of the rate distribution and performed a term enrichment analysis on their associated annotations (S9 Fig). The most significant terms are generally related to DNA repair or extracellular structure, which suggests these processes and components are enriched in proteins whose structural state is rapidly evolving (Fig 2E).

## IDRs have three axes of unconstrained variation

Having calculated the features associated with the sequence segments composing each region in our data set, we then sought to determine if their distributions contained any global structure which would enable us to identify classes with distinct biophysical or functional properties. These distributions are generated by a complex underlying evolutionary process which reflects the combined effects of selection, drift, and mutation. However, to leverage a statistical framework to infer the properties of this process, we fit a Brownian motion (BM) model to each feature calculated from the segments in each region. BM is a simple model of evolution where continuous traits change through a series of small, undirected steps. Thus, the traits accumulate variation at a constant rate over time but do not on average deviate from their original values. BM models are therefore specified by two parameters: a rate ($\sigma^2_{OU}$), which describes the speed at which trait variation accumulates, and a root ($\mu_{\mathrm{BM}}$), which describes the ancestral trait value.

We then applied principal components analysis (PCA) to visualize the major axes of variation of the root and rate parameters for each feature and region. A difficulty with a direct analysis of the parameter estimates, however, is the sensitivity of PCA to differences in scaling between variables, and some features have dramatically different intrinsic scales. For example, many compositional features, like fraction_S, are restricted to the interval [0, 1], whereas SCD is unbounded and can vary from negative to positive infinity. As a result, SCD is responsible for a significant fraction of the overall variance in both parameter distributions (S10–S11 Figs). Therefore, we first normalized the parameters associated with each feature by transforming them into *z*-scores relative to their proteome-wide distributions.

The first two principal components of the root distributions show little overall structure, though there is a slight enrichment of regions along two axes that correlate with acidic and polar features, respectively (S12 Fig). Likewise, the projections of the rates onto the first two principal components are largely distributed along the first (Fig 3A). This observation and the variable amounts of sequence divergence in the regions led us to suspect the first principal component was a measure of the overall rate of sequence evolution. Plotting the first principal component against the sum of the average amino acid and indel rates as measured by substitution models revealed a strong association (Fig 3B). We then projected the rates along second and third principal components to determine if these higher order components contained any additional structure. The resulting distribution is roughly triangular and contains three major axes of variation, corresponding to rapid changes in the regions' proportions of glutamine, charged, and glycine residues (Fig 3C and 3D). Inspection of regions selected along these axes

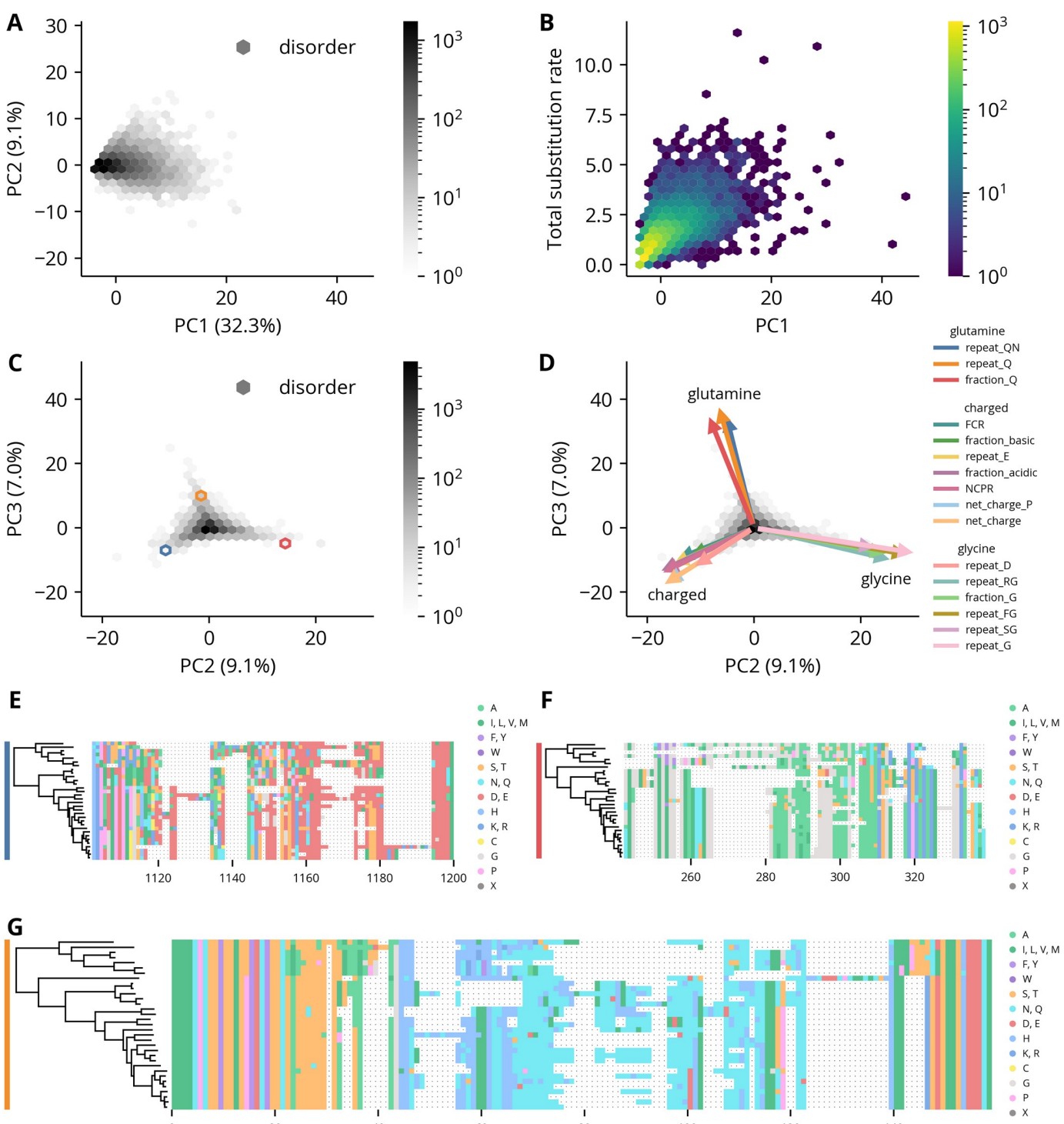

**Fig 3. PCA of disorder regions' feature rates. (A)** The first two PCs of the disorder regions' feature rate distributions. The explained variance percentage of each component is indicated in parentheses. **(B)** Scatter plot of the disorder regions' feature rates along the first PC against the sum of the average amino acid and indel substitution rates in those regions. **(C)** The second and third PCs of the disorder regions' feature rate distributions. The explained variance percentage of each component is indicated in parentheses in the axis labels. **(D)** The same plot as panel C with the projections of original variables onto the components shown as arrows. Only the 16 features with the largest projections are shown. Scaling of the arrows is arbitrary. **(E-G)** Example alignments of disorder regions from the orthologous groups 0A8A, 3139, 04B0, respectively. The colored bars on the left indicate the hexbin containing that region in panel C.

confirmed the high rates of evolution of these features (Fig 3E–3G). Furthermore, we observe a similar distribution when the rates of the order regions were projected along their second and third principal components, which suggests a lack of constraint along these axes is a general property of rapidly evolving proteins (S13 Fig).

### A model of constrained evolution can identify signals of conservation

Though the BM process permits the inference of the rates of feature evolution after accounting for the phylogenetic relationships between species, it does not directly test for their conservation. In fact, under the BM model, trait variation is unconstrained and will increase without bound over time. Instead, evidence of conservation requires comparison to a model where trait variation is constrained. A common choice for modeling the effect of selection on the evolution of a continuous trait is the Ornstein-Uhlenbeck (OU) model. The OU model is similar to the BM model where a trait accumulates variation through a series of small, undirected steps. However, it differs in that the trait is also attracted towards an optimal value where the attraction is proportional to the trait's distance from this value. Under an additional assumption of stationarity that ensures parameter identifiability and estimate consistency, the OU model is accordingly specified with three total parameters: the optimal value ($\mu_{OU}$), the fluctuation magnitude ($\sigma^2_{OU}$), and the selection strength ($\alpha$) [42,43]. While the first two parameters are analogous to the root and rate parameters in the BM model, respectively, the selection strength has no equivalent.

As both models are fully probabilistic, the data's support for the OU model relative to the BM model is quantified by the log ratio of their likelihoods, with greater values indicating more support for the OU model. However, to empirically relate these values to type I and II error rates under a hypothesis testing framework and to assess other statistical properties of the models, we simulated data under each model with a range of values for each of its parameters. Because the likelihood of both models is unchanged if the mean and observations are shifted by a constant, we fixed the mean and optimal values of the BM and OU models to zero in our simulations.

The results show the rate of the BM model is accurately estimated over several orders of magnitude (Fig 4A). Additionally, the variance of these estimates is proportional to the magnitude of the rate, indicated by the constant height of the violin plots in log scale. In contrast, the estimate of the rate of the OU model can have significant bias depending on relative values of the true parameters. For example, in the lower left, when $\log_{10}\left(\frac{\alpha}{\sigma^2_{OU}}\right) < 1$, $\sigma^2_{OU}$ is accurately estimated, as $\alpha$ is small relative to $\sigma^2_{OU}$ (Fig 4B). In the central band, however, when $1 < \log_{10}\left(\frac{\alpha}{\sigma^2_{OU}}\right) < 3$, $\sigma^2_{OU}$, $\sigma^2_{OU}$ is overestimated, as some movement towards the optimal value is likely attributed to a greater fluctuation magnitude. Finally, in the upper right, when $\log_{10}\left(\frac{\alpha}{\sigma^2_{OU}}\right) > 3$, $\sigma^2_{OU}$ is underestimated, as movement from the restoring force overwhelms the contribution of the stochastic component. Similarly, the estimate of the selection strength is biased for many parameter value combinations, though the relationship is more complex (Fig 4C).

We then quantified the type I and II error rates under a hypothesis testing framework. We first calculated the type I error rate as a function of the critical value of the log likelihood ratio for each value of the BM rate separately (Fig 4D). As the overlapping curves indicate the ratios are independent of the true value, we merged the simulations and calculated empirical critical values for 5% (2.58) and 1% (4.20) type I error rates (Fig 4E). Using the 1% critical value, we calculated the type II error rate as a function of the OU parameter values (Fig 4F). The results

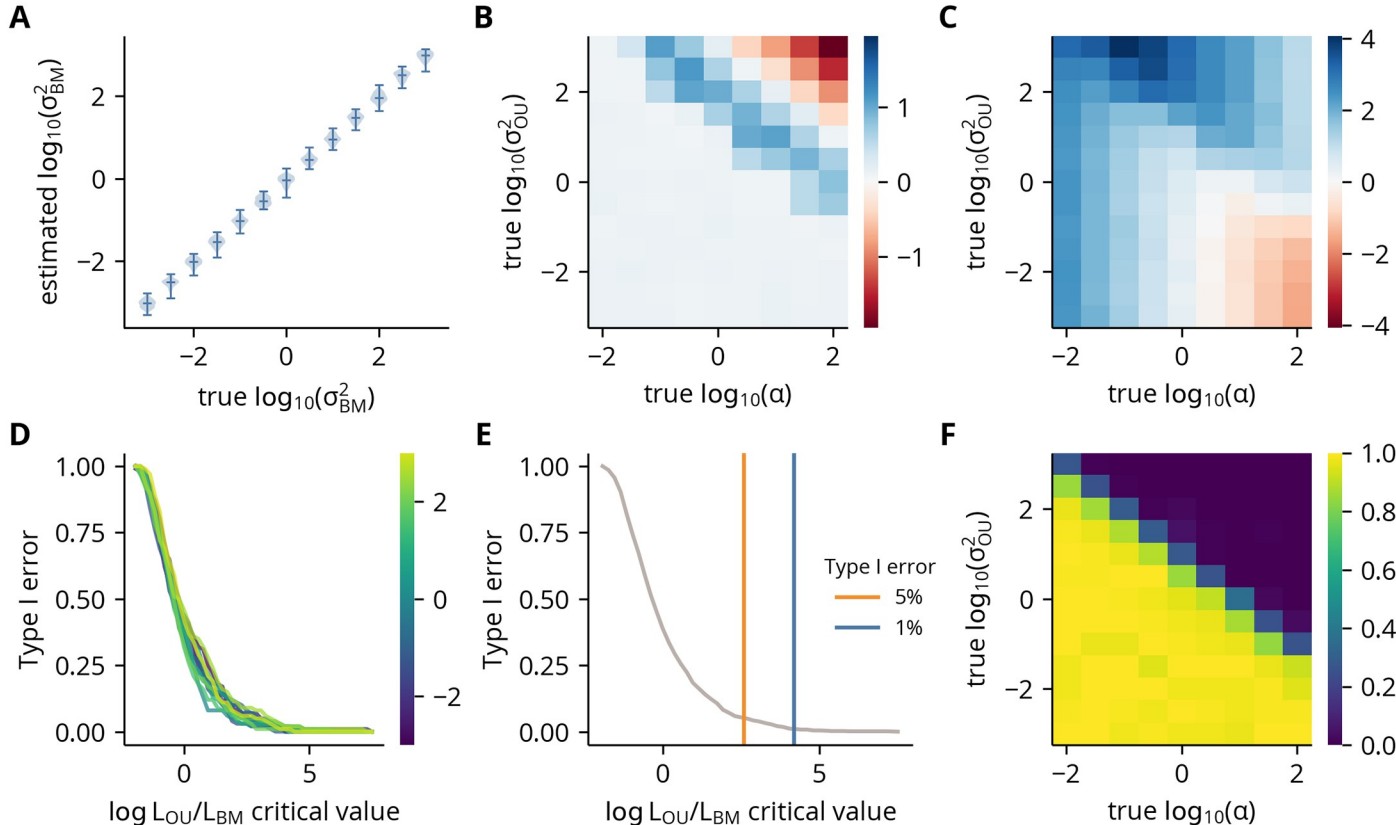

**Fig 4. Hypothesis testing statistics under simulated BM and OU models. (A)** Violin plots of the estimated rates as a function of the true rates ($\sigma^2_{BM}$) under the BM model. **(B)** $\log_{10}$ ratio of the mean estimated rate over its true value as a function of the true rates ($\sigma^2_{OU}$) and selection strengths ($\alpha$). **(C)** $\log_{10}$ ratio of the mean estimated selection strength over its true value as a function of the true rates ($\sigma^2_{OU}$) and selection strengths ($\alpha$). **(D)** The probability of incorrectly rejecting the BM model in favor of the OU model (type I error) as a function of a given critical value of the log likelihood ratio. Each line indicates a different value of the true rate under the BM model. **(E)** Type I error as a function of a given critical value as in panel D but with all simulations under different values of the true rate combined into a single data set. **(F)** The probability of incorrectly failing to reject the BM model in favor of the OU model (type II error) as a function of the true rates ($\sigma^2_{OU}$) and selection strengths ($\alpha$). The probabilities were calculated using the critical value obtained from the empirical 1% type I error rate shown in panel E.

show a strong dependence on the log ratio of $\alpha$ to $\sigma^2_{OU}$, with the error rate sharply declining to zero when $\log_{10}\left(\frac{\alpha}{\sigma^2_{OU}}\right) > 1$. In summary, these results demonstrate that while the parameter estimates of the OU model can have significant bias, when selection is strong relative to the fluctuation magnitude, the log likelihood ratio of the two models can reliably signal against a null hypothesis of Brownian motion.

## Signals of feature conservation are widespread in IDRs

Having evaluated the statistical properties of the hypothesis testing framework, we next sought to investigate the distribution of significant features across all regions. However, to ensure we would not identify signals of conservation simply as a result of high levels of sequence identity, we restricted our subsequent analyses to disorder regions with a minimum amount of divergence as measured by substitution models (S14 Fig). We first quantified the fraction of regions with significant log likelihood ratios using the empirical critical values for 1 and 5% type I error rates, finding that all non-motif features show significance rates that greatly exceed their nominal error rates (Fig 5A). The motif features, though, are more variable, with some

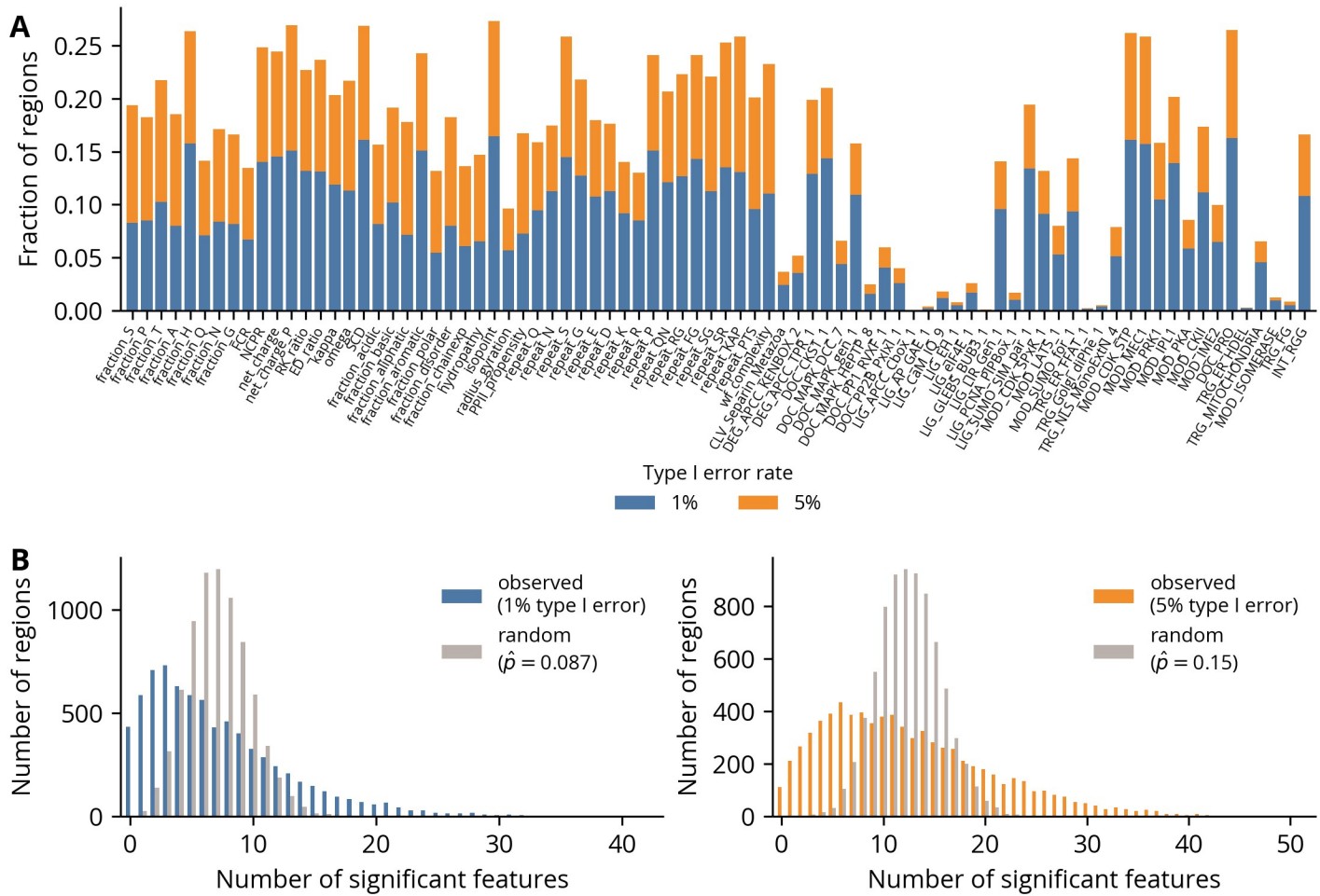

**Fig 5. Distributions of significant features in regions.** (A) Fraction of regions with significant log likelihood ratios for each feature under 1 and 5% type I error rates. (B) Distribution of number of features with significant log likelihood ratios under 1 (left) and 5% type I error rates. The random distributions were computed by randomly shuffling the log likelihood ratios between regions for each feature independently.

reaching significance rates similar to those of the non-motif features, whereas many others have rates at or below the nominal values. Generally, the shorter and more degenerate motifs associated with modification sites have higher rates of significance, which may reflect stronger constraints on motifs which are more likely to arise by random mutations. We next calculated the number of significant features in each region using the same critical values. Both yielded highly skewed distributions, indicating the observation of a significant feature increases the probability of observing another significant feature (Fig 5B). However, as many features have overlapping definitions, this dependence is expected. To correct for this effect, we randomly shuffled the log likelihood ratios between regions for each feature independently and calculated the overall significance rate across all features. For both critical values, the empirical significance rate exceeds the nominal error rate. Thus, constrained feature evolution is widespread in IDRs, with each region containing an average of seven significant features under the 1% type I error rate.

To illustrate the utility of this method for generating hypotheses to guide the functional dissection of specific proteins, we next compared two regions with elevated values of fraction_Q

and repeat_QN but different signals of conservation for each (Fig 6A–6B). Though the alignments of both regions are enriched in glutamine and asparagine residues (Fig 6C and 6E), the first has significant signals of conservation for both features (Fig 6D), whereas the second does not (Fig 6F). Notably, the first region is from an orthologous group that contains a sequence from the *D. melanogaster* gene *ewg*, which is a transcription factor. As polyglutamine stretches can modulate the activity of transcription factor activation domains [44,45], this observation is consistent with the maintenance of an optimal level of transcriptional activation mediated by constraints on glutamine-associated features. In contrast, the low log likelihood ratios of the second region, whose *D. melanogaster* gene is unannotated, are suggestive of unconstrained glutamine repeat expansion and contraction. Interestingly, this region has highly significant log likelihood ratios for several features associated with glycine or phenylalanine repeats despite having few of these residues. However, careful inspection of the alignment reveals several positions where such repeats appear in only one or two sequences, *e.g.*, columns 153 and 208. While these strong signals of conservation may represent artifacts of applying a continuous model of trait evolution to features derived from discrete data, they could also indicate true constraints against the presence of flexible or hydrophobic patches in those sequence segments. Taken together, these results highlight this method's ability to identify biochemical features with signals of constrained evolution while also emphasizing the importance of combining this information with a protein's biological context to jointly suggest testable hypotheses of function.

## Clustering by evolutionary signatures reveals limited patterns of conserved features

Given our observation of widespread signals of feature conservation in *Drosophila* IDRs, we next sought to investigate if there are IDRs with similar patterns of conserved features and common functions. Accordingly, we treated the set of log likelihoods ratios for each region as its "evolutionary signature" and clustered them as in Zarin *et al.* [22]. The resulting heatmap reveals patterns of similar signatures interspersed among a high background of noise, and we identified at least 39 clusters with strong and consistent patterns of constraint (Fig 7). To gain greater insight into their functional properties, we performed term enrichment analyses using the annotations associated with each region's protein. After correcting for multiple comparisons using a 5% false discovery rate, only five clusters were significantly enriched for any terms, which are given in Table 1. Most enriched terms are associated with fewer than ten proteins in each cluster, though the enriched terms for cluster 12 are markedly stronger and consistently associated with protein localization. We also observed that while many clusters are defined by a single strongly conserved feature, three (clusters 15, 23, and 24) have multiple features with signals of constraint. To determine if these more highly constrained regions are associated with a common underlying pattern of features, we compared their evolutionary signatures to normalized values of their inferred optima (Fig 8B–8D). The heatmaps reveal that while the regions in these clusters have common patterns of constraint, the optimal values of their features can vary considerably. Likewise, cluster 12's strongly constrained feature, repeat_K, is similarly variable (Fig 8A). Interestingly, this feature is consistent with its enrichment in annotations related to protein localization, as mitochondrial or nuclear targeting signals are characterized by a net positive charge or clusters of basic residues, respectively [46,47]. However, the relative depletion of lysine repeats in some regions implies they are constrained against the formation of such targeting signals. This interpretation in turn suggests that IDRs may in general have complex relationships between their evolutionary constraints and functions where constraints can both preserve beneficial functions as well as prevent the

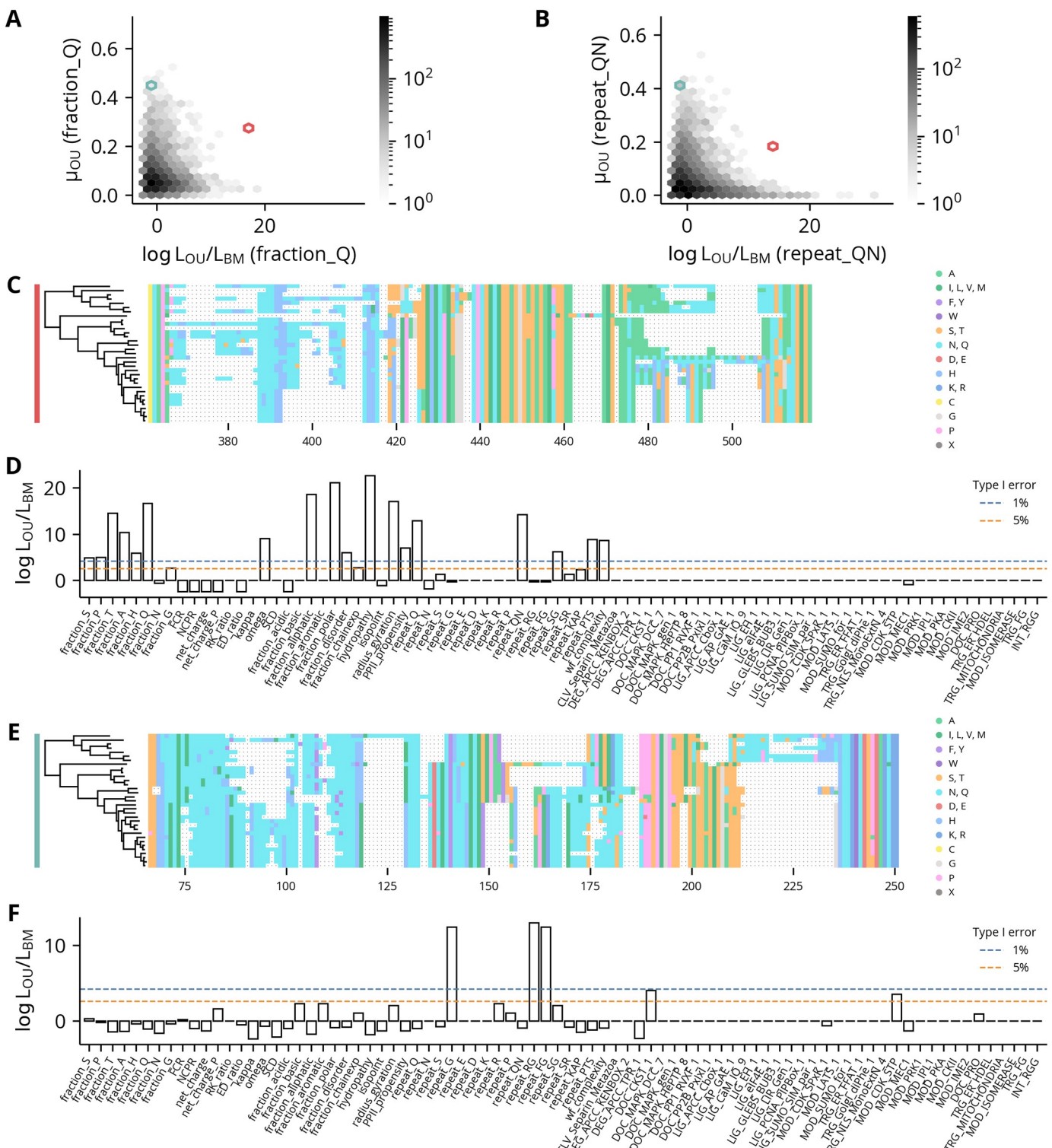

**Fig 6. Example regions with significant and non-significant glutamine-associated features. (A-B)** Scatter plot of the regions' log likelihood ratios against the estimated optimal value for the feature fraction_Q and repeat_QN, respectively. **(C)** Alignment of disorder region from orthologous group 03BB. The colored bar on the left indicates the hexbins containing that region in panels A-B. **(D)** Log likelihood ratios of features for the region in panel C. Critical values for 1 and 5% type I error rates are shown with dotted lines. **(E)** Alignment of disorder region from orthologous group 0715. The colored bar on the left indicates the hexbins containing that region in panels A-B. **(F)** Log likelihood ratios of features for the region in panel E. Critical values for 1 and 5% type I error rates are shown with dotted lines.

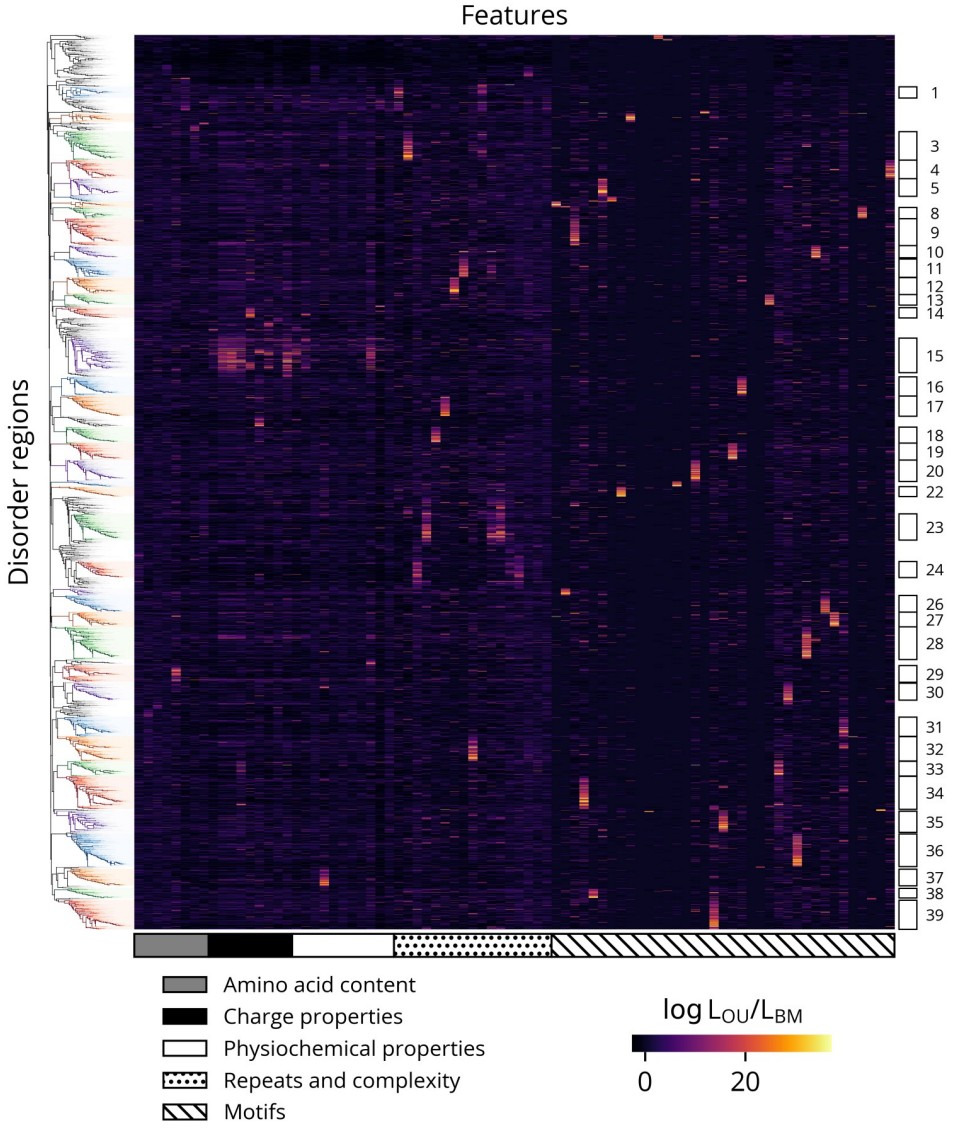

**Fig 7. Hierarchical clustering of evolutionary signatures.** Clusters are indicated by rectangles on the right. Clusters with fewer than 75 regions are omitted for clarity.

acquisition of deleterious ones. Furthermore, though a similar analysis of the inferred optima reveals some analogous patterns of clusters, they are generally more fragmented and diffuse, indicating that despite these caveats, the log likelihood ratios can highlight common patterns of constraint that are not readily apparent from the optimal values of the molecular features alone (S15 Fig).

## Discussion

### IDRs have distinct patterns of sequence and feature evolution

In this study, we applied several phylogenetic models to IDRs to interrogate the evolution of their sequences and molecular features. Most significantly, through a comparison of two

**Table 1. Significantly enriched terms in clusters.**

| Cluster | Regions in cluster | Regions with term | p-value | Corrected p-value | Term ID | Term name |
|---|---|---|---|---|---|---|
| 4 | 158 | 3 | 3.36E-05 | 9.93E-03 | GO:0001671 | ATPase activator activity |
| 4 | 158 | 3 | 3.36E-05 | 9.93E-03 | GO:0140677 | molecular function activator activity |
| 4 | 158 | 7 | 2.46E-04 | 4.84E-02 | GO:0044281 | small molecule metabolic process |
| 6 | 44 | 2 | 1.58E-04 | 4.42E-02 | GO:0044319 | wound healing, spreading of cells |
| 12 | 146 | 15 | 3.05E-05 | 1.65E-02 | GO:0008104 | protein localization |
| 12 | 146 | 16 | 6.42E-05 | 1.74E-02 | GO:0033036 | macromolecule localization |
| 12 | 146 | 8 | 1.51E-04 | 2.73E-02 | GO:0045184 | establishment of protein localization |
| 12 | 146 | 6 | 5.38E-04 | 4.88E-02 | GO:0008233 | peptidase activity |
| 12 | 146 | 7 | 5.03E-04 | 4.88E-02 | GO:0015031 | protein transport |
| 12 | 146 | 5 | 5.73E-04 | 4.88E-02 | GO:0035592 | establishment of protein localization to extracellular region |
| 12 | 146 | 5 | 6.31E-04 | 4.88E-02 | GO:0071692 | protein localization to extracellular region |
| 26 | 141 | 2 | 1.66E-04 | 3.42E-02 | GO:0031543 | peptidyl-proline dioxygenase activity |
| 26 | 141 | 2 | 1.66E-04 | 3.42E-02 | GO:0019511 | peptidyl-proline hydroxylation |
| 26 | 141 | 2 | 1.66E-04 | 3.42E-02 | GO:0018126 | protein hydroxylation |
| 38 | 82 | 4 | 8.30E-05 | 3.78E-02 | GO:0010369 | chromocenter |

models of continuous trait evolution we demonstrate evidence of widespread constraint in IDRs within the *Drosophila* proteome. Furthermore, by quantitatively ranking the importance of various molecular features, these results can generate hypotheses to guide the functional dissection of IDRs in specific proteins. More broadly, though, it suggests that constraint of distributed features is a mechanism of IDR evolution common to multiple biological systems. Using evolutionary signatures derived from these models, we also attempted to identify clusters of IDRs with shared patterns of constraint and common functions. However, in contrast to the previous study in yeast, we found limited evidence of IDR clusters with specific biological functions. This may indicate a more complex regulatory environment in multicellular organisms which lacks a simple mapping between molecular features and functions. For example, as mentioned in the analysis of cluster 12, some constraints may correspond to the absence of a property or function. As annotations are typically framed as positive statements about function, enrichment analyses have less power to detect this type of negative relationship between a feature and its function. Alternatively, conserved features may correspond to classes of conformational ensembles or modes of molecular interaction which are not necessarily associated with specific functions. Technical differences between the two studies may also explain the discrepancy. In particular, the BM and OU processes are highly generic models of trait evolution and may have less power to detect signals of constrained evolution than the simulation approach used in the previous study. The lack of region-specificity in annotations further compounds these issues, as the spurious association of IDRs with annotations derived from other regions introduces noise to the enrichment analyses. However, these and other methodological limitations are discussed in greater detail in a subsequent section.

In addition to the conservation of molecular features, we found IDRs exhibit other distinct patterns of evolution. For example, a comparison of the exchangeability matrices fit to the disorder and order regions shows that, relative to the order regions, the disorder regions have decreased exchangeability coefficients between the disorder-enriched residues. Conversely, the disorder-depleted residues have increased exchangeability coefficients with each other. As a residue's enrichment in IDRs is generally interpreted as a measure of its ability to promote intrinsic disorder, these results indicate that within IDRs and relative to folded domains,

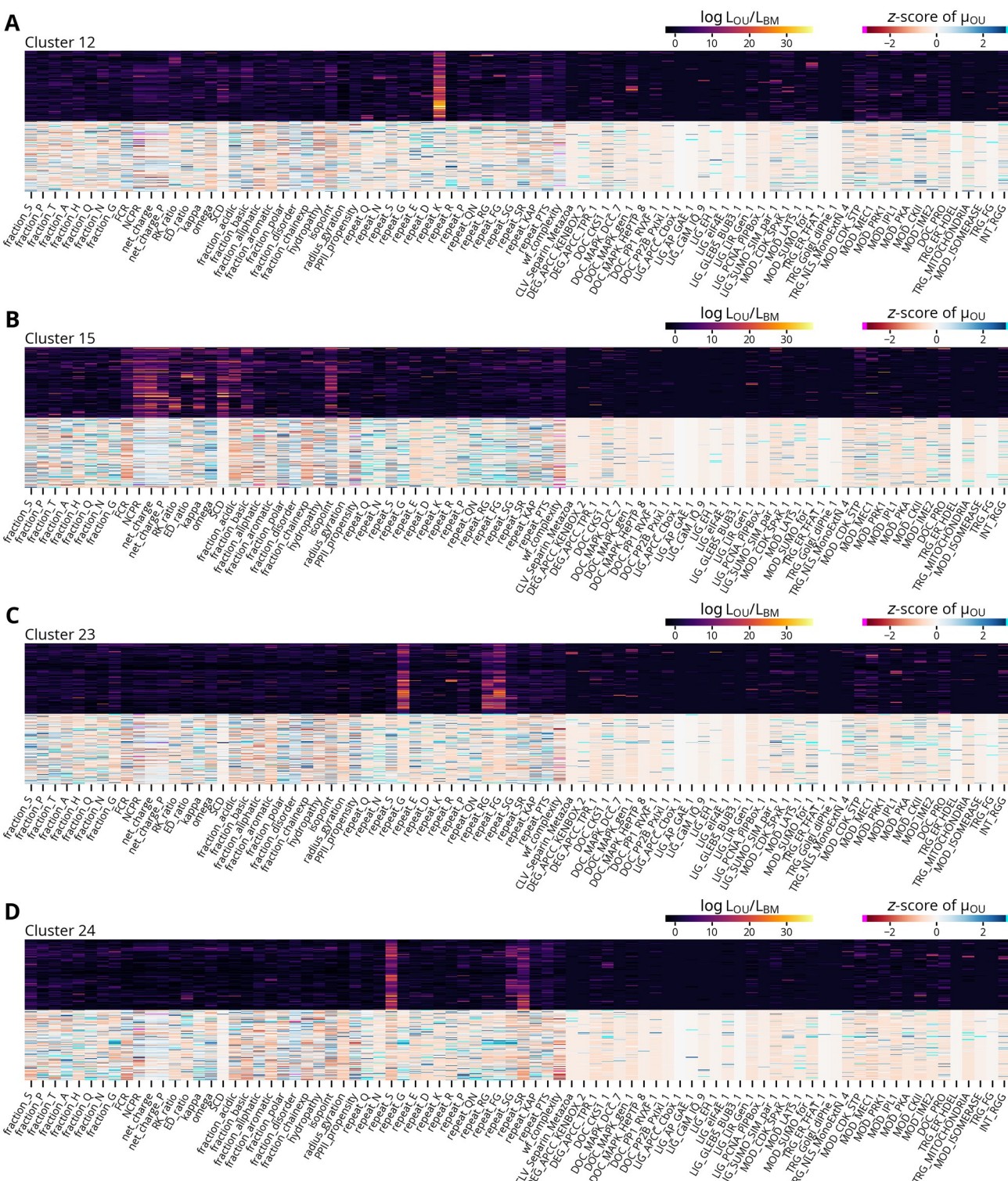

**Fig 8. Example clusters with log likelihood ratios and normalized optimal values. (A-D)** Each panel shows the pattern of log likelihood ratios of the indicated cluster from Fig 7 in the upper half and the corresponding normalized estimated optimal values in the lower half. The optimal values are expressed as z-scores relative to each feature distribution in the subset of rapidly evolving disorder regions clustered in Fig 7. Values below or above negative or positive three are indicated with magenta and cyan, respectively.

disorder-promoting residues are subject to stricter constraints, whereas structure-promoting residues are more biochemically interchangeable. A potential weakness of this analysis is its dependence on sequence alignments which were created using scoring matrices that are in turn derived from other substitution models. Previous studies have attempted to minimize the impact of this circular dependency through an EM-like procedure where substitution models are first fit to alignments, and the sequences in the alignments are then re-aligned with scoring matrices derived from the fit models in alternating rounds until convergence [37,38]. Because our analyses fit the substitution models directly from alignments, the observed patterns are possibly an artifact of using a scoring matrix derived from folded domains to align IDRs. However, as our matrices reproduce the trends reported in these prior studies, they likely reflect true differences in the patterns of residue substitution within IDRs.

This result is a partial reversal of the typical pattern observed in folded proteins where the generally larger and more hydrophobic structure-promoting residues are subject to strict geometric constraints imposed by the tightly packed hydrophobic core. In contrast, the smaller and more hydrophilic disorder-promoting residues are more variable, as they often occur in flexible, solvent-exposed regions. It is, however, consistent with other analyses of sequence-function relationships in IDRs. For example, several studies demonstrated that acidic activation domains of transcription factors, which are usually disordered, contain clusters of hydrophobic residues interspersed throughout their largely acidic chains [48–50]. Additionally, these studies showed that many distinct sequences can yield similar levels of transcriptional activity. Together, these findings suggest a model of transcriptional activation where the repulsions between acidic residues maintain the bulky hydrophobic residues in accessible conformations that in turn allow the activation domains to bind their targets through non-specific hydrophobic interactions. As these interactions do not require highly complementary interfaces, the observed increase in exchangeability coefficients between hydrophobic residues is consistent with this model and may reflect the prevalence of such "fuzzy complexes" in IDR interactions. Furthermore, other studies have shown that different disorder-promoting residues have specific effects on the material properties of condensates formed by phase-separating IDRs. For example, in FUS family proteins glycine residues enhance fluidity, whereas glutamine and serine residues promote hardening [51]. Even glutamine and asparagine residues, which differ by a single methylene group, can have disparate effects on the conformational preferences of IDRs. While glutamine-rich sequences are conformationally heterogeneous and form toxic aggregates, asparagine-rich sequences instead assemble into benign amyloids, as asparagine's shorter side chain promotes the formation of turns and $\beta$-sheets [52]. Thus, these observations along with the decreased exchangeability coefficients between disorder-promoting residues suggest that subtle differences in their biochemical properties may constrain patterns of residue substitution in IDRs.

## Disorder is correlated with many molecular features

Though the mutations generated by evolution are not a random or exhaustive sample of sequence space, they are perturbations of a common ancestor which can reveal the relationship between a region's biophysical properties and its propensity for disorder. Thus, the analysis of feature and score contrasts is effectively a natural "mutational scanning" experiment. We found that disorder scores have the strongest correlations with features that measure a region's overall polarity and hydrophobicity, *e.g.*, fraction_disorder and hydropathy. As the formation of a hydrophobic core is a major driving force in protein folding, this relationship is expected. However, the strength of these correlations indicate that a region's relative proportion of

hydrophilic and hydrophobic residues is, to a first approximation, the largest determinant of predicted intrinsic disorder.

Excluding these hydrophobicity-related features, the next strongest association is a negative correlation with wf_complexity, demonstrating the predictor strongly associates low complexity with intrinsic disorder. However, sequence complexity is a statistical rather than a biophysical criterion, and while many disordered regions have low levels of complexity, some low-complexity regions, like collagen, are structured. This suggests that while disorder predictors are in general accurate classifiers of a residue's structural state [39], they can conflate the correlates of intrinsic disorder with their causes. Therefore, in some cases their predictions may require careful interpretation.

The remaining significant correlations are generally weak and likely reflect a partial redundancy with the more strongly correlated features discussed previously. Interestingly, though, disorder scores are weakly correlated with many motifs, with the signs largely reflecting their class. For example, the correlations with docking (DOC) and ligand binding (LIG) sites are largely negative, whereas those with modification (MOD) sites are generally positive. However, this analysis does not indicate whether the predictor responds to these motifs directly or to features that are correlated with them. For example, docking and ligand binding sites are generally mediated by small hydrophobic patches, so the correlations could reflect an increase in hydrophobicity caused by an additional hydrophobic residue "completing" the motif rather than the motif itself. Likewise, IDRs are highly enriched in phosphorylation sites, many of which are targeted to disorder-promoting residues like serine or threonine. As the disorder scores and features are calculated at the level of regions, whose lengths can exceed 1,000 residues, this analysis is limited in its ability to distinguish these possibilities. However, a more targeted *in silico* mutational analysis would yield further insights.

The GO annotation enrichment analysis indicates that proteins involved in DNA repair and extracellular structures contain a disproportionate number of regions whose disorder scores are rapidly evolving. Because the significance tests were not corrected for multiple testing or controlled for their false discovery rate, we caution against over-interpreting this result and instead consider it as a hypothesis for further investigation. In general, however, the regions with rapidly evolving disorder scores may correspond to molecular recognition features (MoRFs), which are modules in IDRs that undergo a disorder-to-order transition on binding their targets [8,53]. Because MoRFs already exist on the boundary between disorder and structure, small changes in the biophysical properties of these regions may have large effects on their structural ensembles. Thus, the most variable disorder scores may reflect instances where a mutation triggered a "phase transition" between largely structured or disordered native states.

## Future evolutionary analyses of IDRs will require a multimodal approach

As discussed by Zarin *et al.*, the interpretation of evolutionary signatures is complicated by several methodological limitations [22]. For example, because IDRs are identified as contiguous segments of high predicted disorder, their boundaries are defined by adjacent structural elements. This approach can therefore split an IDR that is a single evolutionary or functional unit if it contains a semi-disordered module that scores below the threshold. Conversely, it can also merge two distinct IDRs if they are not separated by at least one folded domain. These issues can have significant effects on the accuracy of our analyses, as mismatches between the inferred boundaries of an IDR and its true evolutionary or functional divisions can introduce noise or even spurious signals of constraint. However, more targeted applications of this

framework where investigators manually choose the boundaries for specific IDRs can ensure the results are directly related to the regions of interest.

Another challenge is the significant overlap in the definitions of the features used in this study. As a result, many are highly correlated, which precludes straightforward quantitative manipulations or interpretations of an IDR's evolutionary signature. Additionally, the features were originally compiled from a variety of previous reports on IDRs. Thus, they likely reflect the biases of individual investigators or highly studied examples rather than constituting a comprehensive set of IDR-associated properties. The original authors have since addressed this in subsequent studies by applying machine learning methods to perform feature selection or learn features directly from alignments of IDRs [23,54]. However, integrating these methods into a unified phylogenetic comparative framework will require further effort.

Despite these caveats, by fitting the BM and OU models to molecular features calculated from alignments of IDRs, we were able to quantify the relative support for constrained and unconstrained models of IDR evolution using a statistical framework. As the BM and OU models describe the evolution of arbitrary continuous traits, a strength of this approach is its independence from assumptions about the underlying process of sequence evolution. In contrast, the previous study used simulations to generate null distributions for a model of no constraint and defined an IDR's evolutionary signature as its deviation from these distributions. However, these comparisons do not directly demonstrate evidence of stabilizing selection but instead test for differences from the null hypothesis. Thus, this approach is highly dependent on the specification and parameterization of these simulated models. Accordingly, an error in either can yield evidence of constraint for an IDR even if none of its molecular features are under selection.

However, the comparative phylogenetics methodology applied here also has limitations. As many features have strict boundaries or cannot vary continuously, they violate one or more of the underlying assumptions of the BM and OU models. Fortunately, for many features these inconsistencies likely do not seriously compromise the analysis. For example, though compositional features like fraction_S are mathematically restricted to the interval between zero and one, they are likely constrained by much narrower selective regimes, and within these regimes, their behavior is effectively described by an OU model. For other features, however, the deviations are more consequential. For example, net_charge and the motif features can only assume integer and natural number values, respectively, which imposes significant restrictions on their allowed increments that are not reflected in the BM and OU models. Instead, more appropriate models for count data are birth-death processes, which are Markov chains defined on the natural numbers. However, though linear birth-death processes are well-studied and widely applied in biology [55], to our knowledge there are no simple parameterizations which describe a mean-reverting behavior analogous to the OU model. Thus, further theoretical developments are needed to apply birth-death processes as a model of stabilizing selection in studies of IDR evolution.

While the BM and OU models are powerful tools for studying trait evolution, their generality limits the specificity of the hypotheses they can test. For example, because the log ratios of the likelihoods are independent from the values of the inferred optima, this analysis cannot distinguish between IDRs which have a common conserved feature but whose values of that feature differ, *e.g.*, IDRs with high and low fractions of glutamine residues. In contrast, because simulation-based approaches can specify arbitrary constraints, they permit investigations of increasingly refined models of IDR evolution. We therefore view the two approaches as complementary and propose the use of hybrid methods where test statistics are derived from phylogenetic comparative methods like the BM model, and simulations generate the null distributions for those test statistics. While simulations can eliminate specific hypotheses, the

substantial resources involved both in designing them and generating samples make exhaustively testing mechanisms of feature constraint by simulation impractical. We instead recommend a workflow that begins with an analysis using general models of trait evolution to suggest specific hypotheses of constraint that are then tested with a simulation-based approach. Another hybrid method involves using sequence permutations to test for the conservation of motifs or patterned features like kappa [56]. In this method, a comparative model like BM provides the test statistic as before, and a sample of shuffled sequences approximates the null distribution. Because the background distribution of residues is preserved, this procedure can specifically test for the conservation of motifs or patterned features in an IDR independent of the conservation of its composition.

However, as IDRs are likely subject to multiple selective pressures where the constraints on different compositional, patterned, or motif features are highly specific to each, we anticipate a range of computational and experimental methods will be needed to disentangle the complex forces driving their evolution. Accordingly, while these results represent a significant step forward in relating sequence to function in IDRs, further studies exploring these and other approaches will undoubtedly reveal new insights into these ubiquitous but poorly understood regions of proteins.

## Materials and methods

### Alignment and species tree provenance

Alignments of 8,566 single copy orthologs and the corresponding outputs of the missing data phylo-HMM were obtained from Singleton *et al.* [33]. Likewise, the LG consensus tree generated by the "non-invariant, 100% redundancy" sampling strategy was used as the input or reference where indicated in subsequent phylogenetic analyses.

### IDR prediction and filtering

Based on its strong performance in a recent assessment of disorder predictors, AUCPreD was chosen to identify regions with a high probability of intrinsic disorder [34,39]. After removing the gap symbols from the sequences in the alignments, the disorder scores of each sequence were predicted individually. (Alignments 0204 and 35C2 contained sequences which exceeded the 10,000-character limit and were excluded from subsequent analyses.) The resulting scores were then aligned using the original alignment. The average score for each position was calculated using Gaussian process sequencing weighting over the LG consensus tree [57]. Any positions inferred as "missing" by the missing data phylo-HMM or to the left or right of the first or last non-gap symbol, respectively, were excluded. For simplicity, the Gaussian process weights were not re-calculated from a tree pruned of the corresponding tips, and instead the weights corresponding to the remaining sequences were re-normalized. The scores at any remaining positions with gap symbols were inferred by linear interpolation from the nearest scored position.

The average disorder scores were converted into contiguous regions with the following method. Two binary masks were defined as positions where the average score exceeded high and low cutoffs of 0.6 and 0.4, respectively. The low-cutoff mask was subjected to an additional binary dilation with a structuring element of size three to merge any contiguous regions separated by a small number of positions with scores below the cutoff. "Seed" regions were then defined as 10 or more contiguous "true" positions in the high-cutoff mask, and "disorder" regions were obtained by expanding the seeds to the left and right until the first "false" position in the low-cutoff mask or the end of the alignment. "Order" regions were taken as the complement of the disorder regions in each alignment.

The regions were filtered with the following criteria. First, segments with non-standard amino acid symbols, which overlapped with any position labeled as "missing" by the missing data phylo-HMM, or whose number of non-gap symbols was below a length cutoff of 30 residues were removed. Regions whose remaining segments failed the set of phylogenetic diversity criteria detailed in S1 Table were excluded. The final set contained 11,445 and 14,927 disorder and order regions, respectively, from 8,466 distinct alignments.

## Calculation of Pfam domain overlaps

The Pfam-A models (version 36.0) were downloaded from InterPro. The *D. melanogaster* sequences were extracted from the alignments and searched against the Pfam models using HMMER 3.4 with a per-domain reporting threshold (option—domE) of $1 \times 10^{-10}$. Overlap fractions for each region were calculated as the number residues overlapping with any Pfam domain hit divided by the total number of residues in that region.

## Fitting substitution models and trees

To fit amino acid substitution matrices to disorder and order regions, 25 meta-alignments for each were constructed by randomly sampling 100,000 columns from the respective regions. To determine the effect of gaps, the maximum fraction of gaps was set at 0, 50, 100%. The combination of the region types and sampling strategy yielded six different sets of meta-alignments. A GTR20 substitution model with four FreeRate categories and optimized state frequencies was fit to each meta-alignment using IQ-TREE 1.6.12 [58]. Exchangeability and rate coefficients were normalized, so the average rate of each model was equal to 1. Because exchangeability and rate coefficients are highly correlated across meta-alignments of the same region type, all figures are derived from the maximum 50% gap fraction meta-alignment sets unless otherwise noted (S3–S5 Figs).

To obtain estimates of the average substitution rates in each region, separate amino acid and indel models were fit to each alignment. For the amino acid substitution models, the columns in the alignments were manually segregated into disorder and order partitions using the regions derived from the AUCPreD scores. However, to prevent poor fits from a lack of data, a partition was created only if it contained a minimum of 20 sequences with at least 30 non-gap symbols. If one partition met these conditions but the other did not, the disallowed partition was consolidated into the allowed one. If neither partition passed, the alignment was skipped. These rules ensured that the regions represented in the final set were fit with substitution models which were concordant with their predicted disorder states. Trees were fit to each partition with an invariant and four discrete gamma rate categories using IQ-TREE 1.6.12 [59]. The disorder partition used a substitution model derived from the average of the state frequencies and exchangeability coefficients fit to the 50% gap fraction meta-alignment sets sampled from the disorder regions. The order partition used the LG substitution model [60]. To prevent overfitting of branch lengths, the trees were restricted to scaled versions of the reference species tree using the—blscale option.

As inference with models that allow insertions and deletions of arbitrary lengths is computationally intractable, a more heuristic approach was taken to quantify the amount of evolutionary divergence resulting from indels in the alignments. For a given alignment, all contiguous subsequences of gap symbols with unique start or stop positions in any sequence were defined as binary characters. Then for each character a sequence was coded with the symbol 1 if the character was contained in that sequence, or it was nested in another contiguous subsequence of gap symbols in that sequence. Otherwise the sequence was coded with the symbol 0. GTR2 models with optimized state frequencies and ascertainment bias corrections were

fit to the resulting character alignments. A discrete gamma rate category was added for every five character columns, up to a maximum of four. To prevent overfitting of branch lengths, the trees were restricted to scaled versions of the reference species tree using the—blscale option.

Because the rate and branch lengths of a phylogenetic substitution model always appear as products in the likelihood expression, they are not jointly identifiable parameters. Instead, the rate is conventionally taken as equal to one (with inverse count units), and the branch lengths are expressed in terms of the expected number of substitution events per column. For models with multiple rate categories, the equivalent condition is that the mean of the prior distribution over the rate categories is equal to one. This effectively makes each rate category a scaling factor of the branch lengths. The inferred rate of a column, calculated as the mean of the posterior distribution over the rate categories, is therefore relative to the average across all columns in the alignment. Thus, an absolute measure of the evolutionary divergence of a column can be obtained by multiplying the inferred rate by the total branch length of the tree. However, as the alignments contain variable numbers of species, this total branch length represents the contribution of both the rate and the tree topology. To normalize for this effect, the total branch length for each tree fit to an alignment was divided by the total branch length of the reference species tree including only the species in that alignment. The reported substitution rate is therefore the product of this scaling factor and the inferred column rate. The average amino acid or indel substitution rate for a region was calculated as the mean of the respective rates across all columns. Because the indel rates were associated with columns in character alignments, they were mapped back to the original sequence alignment by assigning half of a character's rate to its start and stop positions. Since indel models with limited data were prone to overfitting, rates obtained from character alignments with fewer than five columns were set to zero.

## Definition and calculation of features

Features were calculated as in Zarin *et al*. with the following modifications [22]. The regular expression for polar residue fraction was [QNSTCH], which, in contrast to the original study, excludes glycine residues. Additionally, length, expressed in log scale, was replaced with a feature proportional to the radius of gyration for an excluded-volume polymer [61]. Because the radii of gyration of chemically denatured proteins closely match the values expected for equivalent random coils [62], we felt this feature would better capture the relationship between an IDR's length and its biophysical properties. Finally, several motifs from ELM were replaced with their metazoan counterparts or updated versions of the same entries [63]. These differences are noted in the supplementary data. Furthermore, unlike the previous work, motifs were left as counts and not normalized to the proteome-wide average. Kappa, omega, SCD, hydropathy, PPII propensity, and Wootton-Federhen sequence complexity were calculated with local-CIDER 0.1.19 [64]. Isoelectric point was calculated with the Python package isoelectric, which is available on PyPI or at https://isoelectric.org/ [65]. Otherwise, features were implemented with custom code. A full list of features and their definitions is given in S2 and S3 Tables.

## Estimation of Brownian motion and Ornstein-Uhlenbeck parameters

Brownian motion (BM) model parameters for each feature were calculated with two methods. The first used Felsenstein's contrasts algorithm to efficiently calculate roots and contrasts for the disorder scores and features of each region [40,41]. Rates were calculated as the mean of the squares of the contrasts. Though these values are unbiased, they are not maximum likelihood estimates and are inappropriate for use with log likelihood ratio calculations. Thus, they were used for analyses involving only the BM model. For comparison with the Ornstein-Uhlenbeck (OU) model, the BM parameters were calculated by maximizing the likelihood. The OU model

parameters for each feature were also calculated via maximum likelihood estimation as described in Butler *et al.* [66]. To ensure parameter identifiability and estimate consistency, the root was treated as a random variable [42,43]. Thus, the covariance matrix, $V$, was parameterized as $V_{ij} = e^{\alpha d_{ij}}$ where $d_{ij}$ is the tree distance between tips $i$ and $j$, and $\alpha$ is the selection strength [42]. Both the BM and OU models used the reference species tree to parameterize the branch lengths.

## Simulations of BM and OU models

Random variates for both the BM and OU models were generated by calculating the mean vectors and covariance matrices given the simulation parameters and sampling directly from a multivariate normal distribution. The mean vectors were fixed to zero, as the likelihood of each model is unchanged under shifts by a constant. The covariance matrices were calculated using the reference species tree where the branch lengths were scaled by the rate of each model. For the BM model, the rate was varied from $10^{-3}$ to $10^{3}$ in half log steps. For the OU model, the rate and selection strength were varied from $10^{-3}$ to $10^{3}$ and from $10^{-2}$ to $10^{2}$, respectively, both in half log steps. For both models, each parameter combination was simulated 100 times.

## Calculation and clustering of evolutionary signatures

The log likelihoods were calculated for each feature in each region using the maximum likelihood estimates of the parameters for the BM and OU models. The pairwise differences in the log likelihoods of each model yielded a vector with 82 components, each representing the relative goodness of fit of the OU model over the BM model. The vectors were clustered using the correlation distance metric and the UPGMA algorithm. Clusters were manually chosen for subsequent GO analyses. To enrich these clusters for regions with a high likelihood of feature conservation despite low levels of sequence identity, this analysis was restricted to the 7,607 (67%) regions whose amino acid and indel substitution rates exceeded 1 or 0.1, respectively (S14 Fig).

## GO term analyses

The 2022-03-22 go-basic release of the Gene Ontology was obtained from the GO Consortium website [67,68]. The gene association file for the 2022_02 release of the *D. melanogaster* genome annotation was obtained from FlyBase [27]. Obsolete annotations were dropped, and the remaining annotations were filtered by qualifiers and evidence code. The allowed qualifiers were "enables," "contributes_to," "involved_in," "located_in," "part_of," and "is_active_in." The allowed evidence codes were all experimental sources, traceable author statement (TAS), and inferred by curator (IC). The annotations were propagated up the ontology graph and joined with the region sets, so every annotation associated with a gene was associated with the regions derived from that gene. *P*-values were calculated with exact hypergeometric probabilities with regions considered as the sampling unit. For the disorder score analysis, the reference set was the filtered regions, and the enrichment set was the regions in the upper decile of the score rate distribution (S9 Fig). For the cluster analysis, the reference set was the regions after the additional filtering by substitution rates, and the enrichment sets were the regions in each cluster. Additionally, in the cluster analysis, tests were only performed for terms associated with at least two regions, and the set of *p*-values for each cluster were corrected for multiple testing using the Benjamini-Hochberg method.

## Supporting information

**S1 Fig. Summary statistics of disorder and order regions. (A)** Distribution of mean lengths of regions. **(B)** Violin plot of the sums of the average amino acid and indel substitution rates in

the disorder and order regions. The substitution rates are significantly greater in the disorder regions than in the order regions ($p < 1 \times 10^{-10}$, Mann-Whitney $U$ test).
(TIFF)

**S2 Fig. Overlap of disorder and order regions with Pfam domains. (A)** Number of disorder and order regions with zero and non-zero overlap with any Pfam domain. The proportion of regions with no overlap is significantly greater for the disorder regions ($p < 1 \times 10^{-10}$, chi-squared test). **(B)** Histogram of the disorder and order regions' overlaps with any Pfam domain. Regions with zero overlap are excluded to more clearly show the distribution.
(TIFF)

**S3 Fig. Exchangeability matrices fit to meta-alignments yielded by different sampling strategies.** Each panel is a mean of the exchangeability coefficients fit to the meta-alignments yielded by a single sampling strategy ($n = 25$). The prefix and suffix in the title of each panel indicate the maximum gap fraction and region type of the columns in the meta-alignments, respectively. For example, the columns in the "50R_disorder" set of meta-alignments were fewer than 50% gaps and sampled from the disorder regions.
(TIFF)

**S4 Fig. Rate matrices fit to meta-alignments yielded by different sampling strategies.** Each panel is a mean of the rate coefficients fit to the meta-alignments yielded by a single sampling strategy ($n = 25$). See S3 Fig for an explanation of the panel labels.
(TIFF)

**S5 Fig. Correlations between mean exchangeability and rate matrices fit to meta-alignments yielded by different sampling strategies. (A)** Correlations between the mean exchangeability matrices in S3 Fig. **(B)** Correlations between the mean rate matrices in S4 Fig.
(TIFF)

**S6 Fig. Coefficients of variation of the exchangeability matrices.** For all panels, the top and bottoms rows correspond to the 50R_disorder and 50R_order meta-alignment sets, respectively. **(A, D)** Mean exchangeability matrices. **(B, E)** Coefficients of variation (ratio of the standard deviation to the mean) of exchangeability matrices. **(C, F)** The coefficient of variation is inversely proportional to the mean, indicating the variation in the parameter estimates is constant relative to their magnitude.
(TIFF)

**S7 Fig. Coefficients of variation of the rate matrices.** For all panels, the top and bottoms rows correspond to the 50R_disorder and 50R_order meta-alignment sets, respectively. **(A, D)** Mean rate matrices. **(B, E)** Coefficients of variation (ratio of the standard deviation to the mean) of rate matrices. **(C, F)** The coefficient of variation is inversely proportional to the mean, indicating the variation in the parameter estimates is constant relative to their magnitude.
(TIFF)

**S8 Fig. Sample contrasts calculation. (A)** Initial state of tree with three tips. Values of traits at each tip are indicated on the tree and in the table. **(B)** Calculation of first contrast between tips B and C. **(C)** Inference of trait value at internal node A. Its branch length is increased to account for the uncertainty in the estimation of its value. **(D)** Calculation of second contrast between tip D and internal node A.
(TIFF)

**S9 Fig. Histogram of disorder score rates in regions.** The grey interval indicates the upper decile of the distribution across both disorder and order regions, which was used as the input

set for the GO term enrichment analysis.
(TIFF)

**S10 Fig. Variance ratios of disorder regions' feature roots. (A)** Variance ratios before normalization. **(B)** Variance ratios after normalization. **(C)** Scree plot of the explained variance ratio by PC.
(TIFF)

**S11 Fig. Variance ratios of disorder regions' feature rates. (A)** Variance ratios before normalization. **(B)** Variance ratios after normalization. **(C)** Scree plot of the explained variance ratio by PC.
(TIFF)

**S12 Fig. PCA of disorder regions' feature roots. (A)** The first two PCs of the disorder regions' feature root distributions. The explained variance percentage of each component is indicated in parentheses in the axis labels. **(B)** The same plot as panel A with the projections of original variables onto the components shown as arrows. Only the 16 features with the largest projections are shown. Scaling of the arrows is arbitrary.
(TIFF)

**S13 Fig. PCA of order regions' feature rates. (A)** The second and third PCs of the order regions' feature rate distributions. The explained variance percentage of each component is indicated in parentheses in the axis labels. **(B)** The same plot as panel B with the projections of original variables onto the components shown as arrows. Only the 16 features with the largest projections are shown. Scaling of the arrows is arbitrary.
(TIFF)

**S14 Fig. Rate distributions of substitution models fit to disorder regions. (A)** Average amino acid rates in regions. **(B)** Average indel rates in regions. For both panels, the grey intervals correspond to the subsets of rapidly evolving regions used for the clustering and GO term enrichment analyses. 5892 (52%) and 6052 (53%) regions pass the amino acid and indel rate cutoffs, respectively, and 7607 (67%) of regions pass either.
(TIFF)

**S15 Fig. Hierarchical clustering of normalized optimal values.** The optimal values of the OU model ($\mu_{OU}$) are clustered with the same method as the signatures derived from the log likelihood ratios in Fig 7. The optimal values are expressed as $z$-scores relative to each feature distribution in the subset of rapidly evolving disorder regions. Values below or above negative or positive three are indicated with magenta and cyan, respectively.
(TIFF)

**S1 Table. Phylogenetic diversity criteria.**
(XLSX)

**S2 Table. Features and their definitions.**
(XLSX)

**S3 Table. Feature regular expressions.**
(XLSX)

## Acknowledgments

We thank the past and present members of the Eisen lab and Max Staller for helpful discussions and their assistance in revising the manuscript.

## Author Contributions

**Conceptualization:** Marc D. Singleton, Michael B. Eisen.

**Data curation:** Marc D. Singleton.

**Formal analysis:** Marc D. Singleton.

**Funding acquisition:** Michael B. Eisen.

**Investigation:** Marc D. Singleton.

**Methodology:** Marc D. Singleton.

**Resources:** Marc D. Singleton.

**Software:** Marc D. Singleton.

**Supervision:** Michael B. Eisen.

**Visualization:** Marc D. Singleton.

**Writing – original draft:** Marc D. Singleton.

**Writing – review & editing:** Marc D. Singleton.

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
