## [Decision Letter · Decision Letter 0]

11 Feb 2024

Dear Dr. Singleton,

Thank you very much for submitting your manuscript "Evolutionary analyses of IDRs reveal widespread signals of conservation" for consideration at PLOS Computational Biology. As with all papers reviewed by the journal, your manuscript was reviewed by members of the editorial board and by several independent reviewers. The reviewers appreciated the attention to an important topic. Based on the reviews, we are likely to accept this manuscript for publication, providing that you modify the manuscript according to the review recommendations.

Sincerely,

Attila Csikász-Nagy

Academic Editor

PLOS Computational Biology

Zhaolei Zhang

Section Editor

PLOS Computational Biology

Reviewer's Responses to Questions

**Comments to the Authors:**

Reviewer #1: Singleton & Eisen present a thorough evolutionary analysis to reveal conservation of distributed features in IDRs in drosophila using phylogenetic comparative methods. This work is of general interest and is a valuable contribution to the community, building on the pioneering application of such models to gene expression data in Drosophila. The analysis is thorough and well-presented. However, some of the stated conclusions in the text are strong given the underlying evidence and caveats of the study. The limitations of the analyses are quite thoroughly documented in the discussion, but more explicit referral to this section and less overstatement of the evidence (in the main text/pre-discussion) would be helpful to understand the biology and remaining open questions. E.g. there should be a more explicit referral to what I see as the two major caveats of the study: a) that there are artefacts associated with applying continuous models of trait evolution to discrete data, and b) that there is a dependence of the method on substitution matrices inferred from sequence alignments.

Minor comments:

Fig. 2/intrinsic disorder poorly conserved in some proteins:

- there's no explicit mention of the error associated with disorder predictions and how this can impact the variable disorder scores across phylogenies. the authors should rule out the possibility of a technical artefact before claiming that the proteins that contain these IDRs have rapidly evolving structural states. this may especially be the case since the disorder scores can be variable despite high levels of sequence identity (line 172-173).

- though some examples are given in fig. 2, it would be helpful to know, e.g. by the authors' criteria, what proportion of the analyzed IDRs have poorly conserved disorder, and how this compares to previous studies in other species?

Fig. 5/Signals of feature conservation widespread in IDRs

- it would be helpful to know the number of IDRs (/total percentage) that pass the threshold for minimum amount of divergence? Perhaps just an inset in the grey box in Fig S13 or in fig. legend

Fig. 6: although a positive and negative example are presented, the second example in particular does not really serve as validation, since we do not in fact know if the low log likelihood ratios of glutamines in this region are suggestive of unconstrained glutamine repeat expansion/contraction. The fact that it is an unannotated gene does not count as evidence for this. It would be more convincing if the authors could present a validated example, perhaps from a previously studied drosophila IDR.

Fig. 7/clustering analysis: there could be several reasons other than a difference between multi-cellular and uni-cellular IDR evolution for the lack of associated function in clusters:

- given the GO terms in table 1, there seem to be some fairly general terms that are enriched such as "molecular function activator activity" -- if the authors limited the GO analysis to terms that do not apply to most drosophila genes, would they find more specific enrichment?

- the log ratio scores do not contain information about the directionality of the molecular feature conservation (e.g. is it the depletion that's conserved, or the enrichment of that feature?) -- could this decrease the resolution/interpretability of the analysis?

- how does the diversity of included species affect the signal of the assay? is there "enough" divergence? how does it compare to the previous study in yeast? can the authors provide more information on how the drosophila species were chosen in the methods (aside from the reference)?

- how does a clustering of physicochemical features (with no evolutionary information) compare to the results from the OU/BM analysis? Is there any more information gleaned from the phylogenetic modeling (e.g. more enrichment for GO terms in the clusters)?

Reviewer #2: Singleton and Eisen presents a computational study addressing the possible conservation of "evolutionary signatures" in intrinsically disordered protein regions, i.e., general properties as opposed to amino acid sequence. And they find signals of constraint, shaping the evolution of the sequences. However, I must disclose here that I am an experimentalist and not at all an expert in the computational methodology. For a non-expert like me, the results and discussion sections are therefore quite hard to digest, but I suppose this is fine for a journal focussing on computational biology. Nevertheless, the general question addressed in the paper is of high general interest in the field, and so are the results, and this is the relevant feedback that I can give.

Some minor points:

1. How much are the conclusions dependent on a correct prediction of disorder? The authors touches upon this question without addressing it:

"For each residue in the input sequence, AUCPreD returns a score between 0 and 1 where higher values indicate higher confidence in a prediction of intrinsic disorder. In some alignments, the scores vary by nearly this entire range at a given position even when there is a relatively high level of sequence identity"

Even with experimental data at hand the definition of an IDR may not be clear.

2. Title, write out IDR

3. There are a few grammatical errors/missing words here and there.

4. Beginning of Results, line 118; "Although both distributions span several orders of magnitude, the order regions are generally longer than the disorder regions, with means of 105 and 245 residues, respectively." Swap place of 105 and 245.

5. Line 185, isoelectric point

Reviewer #3: In this work, the authors apply several evolutionary analyses to explore conservation in disordered regions. The work appropriately recognizes and builds on prior work from the Moses lab, but introduces several new analyses that are appreciated and interesting. The paper is in general well, written, and most of my comments are aimed solely at making it even clearer to a general audience. I particularly liked the Discussion.

A few of additional papers/preprints the authors may wish to consider in the context of this work (generally) are the following, although none of these are required, they are simply additional literature that I think support the authors conclusions and represent prior relevant work!

Dasmeh, P., Doronin, R. & Wagner, A. The length scale of multivalent interactions is evolutionarily conserved in fungal and vertebrate phase-separating proteins. Genetics 220, (2022).

Cohan, M. C., Shinn, M. K., Lalmansingh, J. M. & Pappu, R. V. Uncovering Non-random Binary Patterns Within Sequences of Intrinsically Disordered Proteins. J. Mol. Biol. 434, 167373 (2022).

Langstein-Skora, I., Schmid, A. & Emenecker, R. J. Sequence-and chemical specificity define the functional landscape of intrinsically disordered regions. BioRxiv (2022). at <https: 10.1101="" 2022.02.10.480018.abstract="" content="" www.biorxiv.org="">

Forcelloni, S. & Giansanti, A. Evolutionary Forces and Codon Bias in Different Flavors of Intrinsic Disorder in the Human Proteome. J. Mol. Evol. 88, 164–178 (2020).

Brown, C. J., Johnson, A. K., Dunker, A. K. & Daughdrill, G. W. Evolution and disorder. Curr. Opin. Struct. Biol. 21, 441–446 (2011).

There are a million things I could ask the authors to do, of course, but these would be subjective, and I’m not convinced they would really improve the paper (or at least to an extent that would be worth the additional time taken). With that in mind, my comments are exclusively focused on ensuring this work is clearly presented.

Finally, the authors have done an excellent job ensuring their code is fully public and accessible.

Minor comments:

Line 109

“We identified regions with high levels of inferred intrinsic disorder in over 8,500 high quality alignments of single copy orthologs from 33 species in the Drosophila genus using the disorder predictor AUCPreD”.

If these alignments were pre-existing, it would be useful to briefly expand on where they came from and how they were generated. If they were created for this study, it would similarly be useful to expand on how they were generated. The fact these are single-copy orthologs is important, and it would be good for the authors to briefly expand on why this is the case and how this was ensured.

Line 119:

“We then quantified the sequence divergence in each region by fitting phylogenetic trees to the alignments using amino acid and indel substitution models.”

It would be useful to add one more sentence (and a reference or two?) explaining what this does and how it works (for a less expert audience).

Line 150

To what extent are the exchangeability matrices in Fig. 1 really the same, and differences are solely determined by differences in composition for the disordered/ordered regions? It seems - by eye, anyway - that, as stated in the text, the patterns are very similar, and while the log10 ratios show differences the major differences appear to be associated with the rarest amino acids, which I tend to think maybe noise more than a bona fide signal? For example, do we really think C->E/D is meaningfully different or is the log10 ratio of two very small numbers inherently sensitive to small fluctuations because the absolute values are small? To be clear, this is not a problem, but I guess I’m unconvinced there is strong evidence for meaningful differences in exchangeability coefficients for residues in disordered vs. ordered regions (which is actually really interesting!).

Line 180

I 100% agree with the authors stated concern re: comparing evolutionarily related sequences, but the description of what the ‘contrasts’ are here left me unsure of what exactly is going on. I’d appreciate it if the authors could expand a bit on this, perhaps even with a schematized figure explaining conceptually how this works (specifically expanding on the "general assumptions of the underlying evolutionary process, are independent and identically-distributed" [everything up until here is very clear]). It would also be helpful to consistently refer to ‘disorder scores’ as ‘disorder scores’ instead of alternating between ‘disorder scores’ and just ‘scores’.

Line 206

I think it would be extremely instructive to provide a short additional paragraph explaining algorithmically how the BM model is implemented. For example, is this done by taking a region, and then making small random sequence perturbations at the level of single amino acid substitutions at a rate defined by σ^2_{OU} where the substitutions are accepted/rejected based on the resulting deviation from μ_{bm}? Regardless, it would be useful to in just a few sentences explain the implementation here. Moreover, it wasn’t clear to me if this process is done focussing on a single feature (i.e. μ_{bm} for a single sequence property) or if this is some kind of random walk through a high dimensional space. Again, I think this would be clarified by a few sentences just outlining the mechanics of how the BM model is implemented.

Line 245

Same comment as above here for the OU model, written ideally in a way that provides a clear comparison with the BM model.

Figures

For Fig S1. it might be useful to place y-axis on a log scale to make the distributions more visually accessible (this is just a suggestion, and the authors may disagree, which is fine!).

Fig 1.

Please label the matrices so we can understand what is being shown without needing to cross-reference against the figure caption.

Fig 3.D

Relating the many overlapping lines to the many similar colors here is very tricky. I suggest breaking the legend down and just moving each of the sequence feature terms to sit alongside the associated arrow so we, the reader, can do zero work and easily interpret which arrow reports on which feature.

A general comment on the figures:

One thing to consider is whether a reader could look at the figures in the absence of the captions and text and make sense of what was being shown? In many cases, I would suggest probably not. This is perhaps a personal preference as opposed to a hard-and-fast rule, but the authors may wish to consider this question if/as they revise figures</https:>

**Have the authors made all data and (if applicable) computational code underlying the findings in their manuscript fully available?**

Reviewer #1: Yes

Reviewer #2:&nbs

---

## [Decision Letter · Decision Letter 1]

28 Mar 2024

Dear Dr. Singleton,

We are pleased to inform you that your manuscript 'Evolutionary analyses of intrinsically disordered regions reveal widespread signals of conservation' has been provisionally accepted for publication in PLOS Computational Biology.

Best regards,

Attila Csikász-Nagy

Academic Editor

PLOS Computational Biology

Zhaolei Zhang

Section Editor

PLOS Computational Biology

Reviewer's Responses to Questions

**Comments to the Authors:**

Reviewer #1: Thank you to the authors for comprehensively addressing all of the comments, they have addressed all of my suggestions. The only minor/completely optional followup that I have is that a relevant study on the human IDR-ome was pre-printed recently which they may want to cite/include in their manuscript (Pritisanac et al, "A Functional Map of the Human Intrinsically Disordered Proteome").

Reviewer #3: The authors have done a fine job addressing the reviewers' various questions.

It's worth mentioning if the AUCPred predictions continue to raise concerns for other reviewers, metapredict is an alternative (https://metapredict.net/), which has a colab notebook that does proteome-scale FASTA file-derived predictions in <60 seconds and has comparable accuracy to the state of the art. In any case, this is nice work and I fully support publication!

**Have the authors made all data and (if applicable) computational code underlying the findings in their manuscript fully available?**

Reviewer #1: Yes

Reviewer #3: Yes

PLOS authors have the option to publish the peer review history of their article (what does this mean?). If published, this will include your full peer review and any attached files.

Reviewer #1: No

Reviewer #3: No

---

## [Editor Report · Acceptance letter]

9 Apr 2024

PCOMPBIOL-D-24-00042R1 

Evolutionary analyses of intrinsically disordered regions reveal widespread signals of conservation

Dear Dr Singleton,

I am pleased to inform you that your manuscript has been formally accepted for publication in PLOS Computational Biology. Your manuscript is now with our production department and you will be notified of the publication date in due course.

With kind regards,

Judit Kozma
